# Observational study comparing heart rate in crying and non-crying but breathing infants at birth

Antti Juhani Kukka ![ORCID] ,[1,2] Sara K Berkelhamer,[3] Joar Eilevstjønn ![ORCID] ,[4] Thomas Ragnar Wood,[3,5] Omkar Basnet,[6] Ashish KC ![ORCID] [1,7]

AJK and SKB contributed equally.

AJK and SKB are joint first authors.

For numbered affiliations see end of article.

**Correspondence to**
Dr Antti Juhani Kukka; antti. kukka@kbh.uu.se

## ABSTRACT

**Background** Stimulating infants to elicit a cry at birth is common but could result in unnecessary handling. We evaluated heart rate in infants who were crying versus non-crying but breathing immediately after birth.

**Methods** This was single-centre observational study of singleton, vaginally born infants at ≥33 weeks of gestation. Infants who were *crying* or *non-crying but breathing* within 30 s after birth were included. Background demographic data and delivery room events were recorded using tablet-based applications and synchronised with continuous heart rate data recorded by a dry-electrode electrocardiographic monitor. Heart rate centile curves for the first 3 min of life were generated with piecewise regression analysis. Odds of bradycardia and tachycardia were compared using multiple logistic regression.

**Results** 1155 crying and 54 non-crying but breathing neonates were included in the final analyses. There were no significant differences in the demographic and obstetric factors between the cohorts. Non-crying but breathing infants had higher rates of early cord clamping <60 s after birth (75.9% vs 46.5%) and admission to the neonatal intensive care unit (13.0% vs 4.3%). There were no significant differences in median heart rates between the cohorts. Non-crying but breathing infants had higher odds of bradycardia (heart rate <100 beats/min, adjusted OR 2.64, 95% CI 1.34 to 5.17) and tachycardia (heart rate ≥200 beats/min, adjusted OR 2.86, 95% CI 1.50 to 5.47).

**Conclusion** Infants who are quietly breathing but do not cry after birth have an increased risk of both bradycardia and tachycardia, and admission to the neonatal intensive care unit.

**Trial registration number** ISRCTN18148368.

## WHAT IS ALREADY KNOWN ON THIS TOPIC

⇒ Stimulation of breathing infants to elicit a cry is a common practice in newborn deliveries. However, international resuscitation algorithms disagree on whether *crying* or *breathing* should guide providing stimulation.

## WHAT THIS STUDY ADDS

⇒ Non-crying but breathing infants have increased risk of bradycardia, tachycardia and neonatal intensive care admission compared with crying infants.

## HOW THIS STUDY MIGHT AFFECT RESEARCH, PRACTICE OR POLICY

⇒ While waiting for further studies assessing the role of stimulation in non-crying but breathing infants, these newborns should be monitored carefully for clinical deterioration.

to direct further actions in newborn resuscitation.[2]

Lung fluids play a critical role in early fetal lung development but represent an obstacle to lung expansion and gas exchange post delivery.[3 4] Airway liquid clearance is primarily driven by the expansion of the chest wall during inspiration.[5] As compared with quiet tidal breathing, crying similarly to grunting creates greater peak inspiratory flow and a more complex biphasic expiratory volume pattern preventing backflow of liquid into the alveoli.[5 6] This pattern is associated with optimal volume redistribution from well-aerated regions to those which are not as well aerated via pendelluft flow.[5] Crying is thus better suited to both rapid aeration and maintenance of functional residual capacity than tidal breathing.[5]

Lung recruitment at birth also plays a key role in cardiovascular transition and pulmonary blood flow. Lung aeration increases arterial oxygen content with subsequent decrease in local pulmonary vascular resistance, while shift of liquid from the alveoli to perialveolar tissue triggers a neurally mediated global

## INTRODUCTION

The transition to air breathing at birth is a seminal physiological event essential to the onset of pulmonary gas exchange and postnatal survival.[1] This transition is usually accompanied by a spontaneous or elicited cry, a reassuring and readily observable sign for both healthcare providers and family. The clinical implication of a breathing infant who *does not cry* during the first minute after birth warrants evaluation as the globally important Helping Babies Breathe (HBB) neonatal resuscitation algorithm uses lack of cry alone

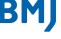

increase in pulmonary blood flow.[7] This transition is key to supporting cardiac output with increases in pulmonary venous return.

With clamping of the umbilical cord and closure of right-to-left shunts, pulmonary venous return becomes the sole source of left intracardiac volume.[7] If the cord is clamped before the lung is aerated, clinical compromise and a fall in cardiac output can theoretically lead to bradycardia.[4 8] However, a trial of physiologically based cord clamping following established lung aeration did not show a reduction in the rate of bradycardia in infants needing resuscitation when compared with early cord clamping.[9]

For low-resource settings, simple clinical signs are needed for identifying infants in need of resuscitation. We previously demonstrated higher risk of neonatal mortality among neonates who were not crying immediately after birth.[10] This observation was particularly true for non-breathing infants, but even non-crying but breathing newborns had an increased risk of predischarge mortality when compared with crying neonates (adjusted Odds Ratio (aOR) 12.3, 95% CI, 5.8 to 26.1).[10] To further investigate the contribution of cry to the transitional physiology of breathing infants, we sought to compare the HR pattern in the first 3 min in infants classified as *crying* or *non-crying but breathing* immediately after birth.

## METHODS
### Design
This was a secondary analysis of a single-centre prospective observational study conducted over months (28 May–28 October 2019) at Western Regional Hospital, Pokhara, Nepal.[11] The hospital has approximately 6500 births annually and is the public referral hospital of the province.

Deliveries were attended by a nurse-midwife or a skilled birth attendant with on-call support from a general medical doctor, paediatrician and obstetrician when needed.[12] Providers were trained in Rapid Feedback for quality Improvement in Neonatal rEsuscitation (REFINE), an HR-guided neonatal resuscitation protocol, including HBB second edition supplemented by a half-day orientation on the use of continuous HR monitoring.[12]

### Participants
All live-born vaginally delivered singleton neonates at or beyond 33 weeks of gestation or with birth weight of >2500 g were eligible for this study.

Crying newborns were defined as those who cried within approximately 30 s after birth.

Non-crying but breathing newborns were defined as those who did not cry but were noted to be breathing within the same period.

All newborns who were non-crying and non-breathing after birth were excluded from this study.

### Data collection and management
Research nurses not part of the clinical care team recorded delivery room observations including periods of breathing and time of cord clamping using a purpose-designed application (Liveborn, Laerdal Medical, Stavanger, Norway).[12] Oxygen saturation data were not recorded. Crying status after birth, and maternal and neonatal demographics were recorded on a supplemental application (SUSTAIN). Cases where breathing status recorded on Liveborn application did not align with the crying status documented on SUSTAIN were reviewed by an expert panel (AKC, SKB, JE and OB) blinded to the HR data. Comments provided by research nurses and actions taken during resuscitation were taken into consideration. As a result of the review, four cases were reclassified from non-breathing to non-crying but breathing and 18 cases were changed to non-breathing and thereby excluded.

Following thorough drying, a NeoBeat newborn HR meter (Laerdal Global Health, Stavanger, Norway) was placed on the abdomen of the newborn.[13] NeoBeat uses a proprietary algorithm based on zero crossing count that averages HR over up to 12 RR intervals. The HR data from NeoBeat was streamed to and time-synced with data collected on the Liveborn application. Bluetooth connection issues or excessive movement of the newborn resulting in absent or highly inconsistent HR required exclusion of cases.

Information from the Liveborn and SUSTAIN apps were extracted separately and cleaned. Unique identification numbers provided for each participant facilitated synchronisation of data for final analysis.

### Data analysis
#### Demographics
For the final data analysis, we excluded participants' data when the birth was registered in the Liveborn app more than 10 s after the actual birth; time of birth was not provided; or if there was too little (<30 s), too late (>60 s after birth) or too inconsistent HR data.

Maternal and infant demographics, mode of delivery, birth weight, small for gestational age status (birth weight <10th centile for gestational age), gestational age, sex and timing of cord clamping (early <60 s and late ≥60 s after birth) were collected from patient records. A list of maternal and obstetric complications considered is presented in online supplemental table 1.

Means in the two cohorts were compared using independent samples t-test and medians by non-parametric median test. Proportions were compared by Pearson's $\chi^2$ test.

#### Heart rate data
HR data for the crying and non-crying but breathing groups from 10 s until 180 s after birth were analysed using centile curves at 3rd, 10th, 25th, 50th, 75th, 90th and 97th centiles. The curves were smoothed using piecewise cubic spline fitting. Median HRs at two serial time

points were compared every 15 s using Mann-Whitney U test with post hoc Bonferroni correction of p values to adjust for repeated measures (p=0.05/12=p≤0.004 for significance).

The ORs of bradycardia and tachycardia were explored by logistic regression. Time of cord clamping, gestational age in weeks, birth weight in kilogram and obstetric complications including abnormal fetal HR were included as covariates. Results are presented as aOR with 95% CI.

Cumulative proportion of neonates with bradycardia (HR <100 beats/min for ≥1 s) and tachycardia (HR ≥200 beats/min for ≥1 s) in the crying and non-crying but breathing groups were presented using a generalised additive model with 95% CI of continuous data recorded every second.

Analyses were performed using R V.4.1.2 (Vienna, Austria) and MATLAB R2021a (MathWorks, Natick, Massachusetts, USA).

### Patient and public involvement

No patient was involved in this study.

### RESULTS

During the study period, 3578 deliveries took place at the hospital, of which 1028 were not assessed for eligibility. Of the 2550 births observed, Liveborn was not used in 368 neonates; 57 resulted in a fresh stillbirth and 34 neonates had a gestational age less than 33 weeks. Of the neonates observed with Liveborn, 882 were excluded due to issues with the HR data, Liveborn application or gestational age. Among the 1209 neonates included in the final analyses, 1155 were classified as crying and 54 as non-crying but breathing immediately after birth (figure 1).

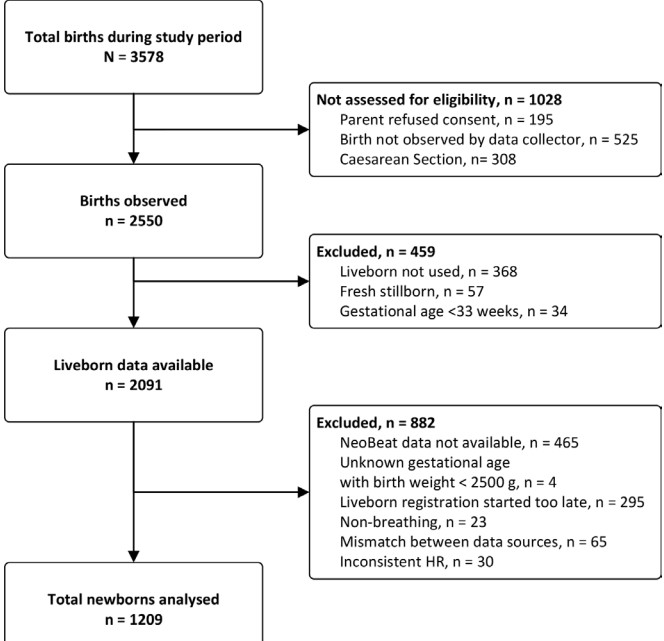

**Figure 1** Flow diagram of stydy inclusion

There was no significant difference in maternal age, parity or ethnicity in the two cohorts. Infants in the two groups had similar birth weight, gestational age, sex distribution and median Apgar scores at 1 and 5 min (table 1). Obstetric complications during admission, abnormal fetal HR during labour and rates of instrumental delivery were also comparable (online supplemental table 1).

Non-crying but breathing infants had a higher proportion of early cord clamping than crying infants (75.9% vs 46.5%, p<0.001) and were more often admitted to the neonatal intensive care unit (13.0% vs 4.3%, p=0.003) (table 2).

The smoothed 3rd, 10th, 25th, 50th, 75th, 90th and 97th HR centiles in crying and non-crying but breathing groups are displayed in figure 2A,B. The 10th centile HR remained <100 beats/min out to 35 and 20 s in the non-crying but breathing and crying cohorts, respectively. There were no significant differences in median HRs in the crying and non-crying but breathing cohorts between 10 s and 180 s (online supplemental table 2).

The aOR of bradycardia during the first 180 s after birth was increased in the non-crying but breathing cohort compared with the crying cohort (aOR 2.64, 95% CI 1.34 to 5.17; p=0.005). Similarly, the aOR of tachycardia was increased to 2.86 (95% CI 1.50 to 5.47, p=0.002) (table 3).

The median duration of the longest episode of bradycardia was 7.5 s (interquartile range [IQR] 3-16 s) and that of tachycardia 10 s (IQR 5-25 s). A total of 12/54 (22.2%) and 112/1155 (9.7%) infants were ever bradycardic by 180 s after birth, and 14/54 (25.9%) and 125/1155 (10.8%) were ever tachycardic in the non-crying and crying groups, respectively. Bradycardia and tachycardia largely affected separate infants; 10 infants were both bradycardic and tachycardic within the first 180 s after birth.

A comparison of the cumulative proportion of infants with bradycardia and tachycardia in the crying and non-crying but breathing groups is presented figure 3A,B. Bradycardia appeared to be a relatively early phenomenon, with tachycardia appearing later, particularly in the non-crying but breathing group.

### DISCUSSION

Crying confers physiological advantage over quiet breathing for clearance of lung liquids at birth. This single-centre observational study compared HR in 1155 crying and 54 non-crying but breathing neonates during the first 3 min after birth. There was no difference in the median HR between the two groups, but a larger proportion of non-crying but breathing neonates displayed an unstable HR pattern of either bradycardia (aOR 2.64, 95% CI 1.34 to 5.17; p=0.005) or tachycardia (aOR 2.86, 95% CI 1.50 to 5.47; p=0.002). Non-crying but breathing infants also had a higher risk of admission to the neonatal intensive care unit.

Previous papers have presented centile curves for neonatal HR.[9 14–16] Crying infants were similar to term

**Table 1** Demographic, obstetric and neonatal characteristics of crying and non-crying but breathing infants

| Indicators | Crying (N=1155) | Non-crying but breathing (N=54) | P value |
|---|---|---|---|
| Demographic factors | | | |
| Maternal age, mean±SD | 24.2±4.3 | 24.1±4.6 | 0.29 |
| Parity, n (%) | | | 0.30 |
| Primipara | 622 (53.9) | 33 (61.1) | |
| Multipara | 533 (46.1) | 21 (38.9) | |
| Ethnicity, n (%) | | | 0.45 |
| Dalit | 250 (21.6) | 15 (27.8) | |
| Janajati | 443 (38.4) | 24 (44.4) | |
| Madheshi | 13 (1.1) | 0 (0.0) | |
| Muslim | 24 (2.1) | 1 (1.9) | |
| Brahmin/Chhetri | 425 (36.8) | 14 (25.9) | |
| Others | 0 (0.0) | 1 (0.9) | |
| Pregnancy and delivery | | | |
| Obstetric complications, n (%) | | | 0.09 |
| No | 1119 (96.9) | 50 (92.6) | |
| Yes | 36 (3.1) | 4 (7.4) | |
| Abnormal fetal HR during labour, n (%) | | | 0.47 |
| No | 1144 (99.0) | 54 (100.0) | |
| Yes | 11 (1.0) | 0 (0.0) | |
| Mode of delivery, n (%) | | | 0.13 |
| Spontaneous vaginal | 1115 (96.5) | 50 (92.6) | |
| Instrumental | 40 (3.5) | 4 (7.4) | |
| Neonatal factors | | | |
| Birth weight (g),* mean±SD | 3065.7±414.8 | 3047.2±410.5 | 0.55 |
| Gestational age (weeks),† mean±SD | 39.2±1.3 | 39.1±1.6 | 0.19 |
| Preterm <37 weeks,† n (%) | | | 0.56 |
| No | 1095 (96.1) | 51 (94.4) | |
| Yes | 45 (3.9) | 3 (5.6) | |
| Small for gestational age <10th centile,‡ n (%) | | | 0.66 |
| No | 942 (82.8) | 46 (85.2) | |
| Yes | 195 (17.2) | 8 (14.8) | |
| Cord clamping, n (%) | | | <0.001 |
| Early <60 s after birth | 537 (46.5) | 41 (75.9) | |
| Delayed ≥60 s after birth | 618 (53.5) | 13 (24.1) | |
| Sex, n (%) | | | 0.86 |
| Male | 634 (54.9) | 29 (53.7) | |
| Female | 521 (45.1) | 25 (46.3) | |
| Apgar score, median (IQR) | | | |
| At 1 min | 7 (7–7) | 6 (6–6) | 0.46 |
| At 5 min | 8 (8–8) | 7 (7–8) | 0.68 |

Missing *=3, †15, ‡19.
$\chi^2$ test, independent samples t-test, non-parametric comparison of medians.
HR = heart rate, IQR = interquartile range, SD = standard deviation

**Table 2** Care of crying and non-crying but breathing infants

| Indicators | Crying (N=1155) | Non-crying but breathing (N=54) | P value |
|---|---|---|---|
| Positive pressure ventilation <3 min, n (%) | | | 0.76 |
| No | 1153 (99.8) | 54 (100.0) | |
| Yes | 2 (0.2) | 0 (0.0) | |
| Transfer to neonatal intensive care unit, n (%) | | | 0.003 |
| No | 1105 (95.7) | 47 (87.0) | |
| Yes | 50 (4.3) | 7 (13.0) | |

χ2 test.

vaginally born cohort described by Bjorland *et al*,[15] but non-crying but breathing infants were more likely to remain bradycardic and experience delayed tachycardia. Prior studies have identified an increased risk of bradycardia among preterm neonates[14]; however, the distribution of gestational age and birth weight did not differ between the crying and non-crying but breathing groups

in our study. Timing of cord clamping also impacts HR, with early clamping increasing the risk of bradycardia.[16 17] In this study, non-crying but breathing neonates were more likely to undergo early cord clamping than crying neonates (75.9% vs 46.5%), but the difference in odds of bradycardia remained statistically significant after this and other adjustments. Early cord clamping may, however, explain some of the differences in timing of bradycardia observed. Furthermore, the clinicians' decision to perform early cord clamping might have been impacted by the lack of early cry or the observed HR.

Another cause of bradycardia in newborns is hypoxia-induced vagal reflex.[18] This phenomenon has been shown most convincingly in relation to apnoea in the setting of obstructed placental blood flow,[19] but as newborns who do not cry at birth have slower clearance of liquid from airways,[5] it is conceivable that even non-crying but breathing neonates could have lower oxygenation than crying neonates. Unfortunately, we were not able to measure peripheral saturation during the study to verify this potential explanatory mechanism. Lack of pulse oximeters is a common limitation in low-income settings, and until their availability increases, caregivers need actionable clinical signs like crying to guide their resuscitation efforts.

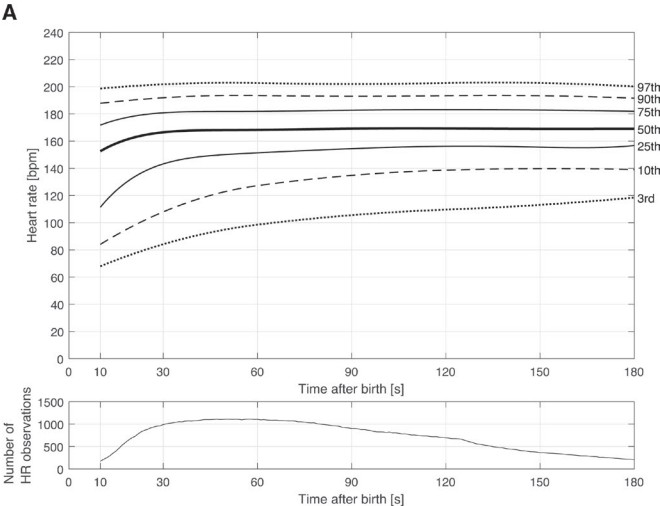

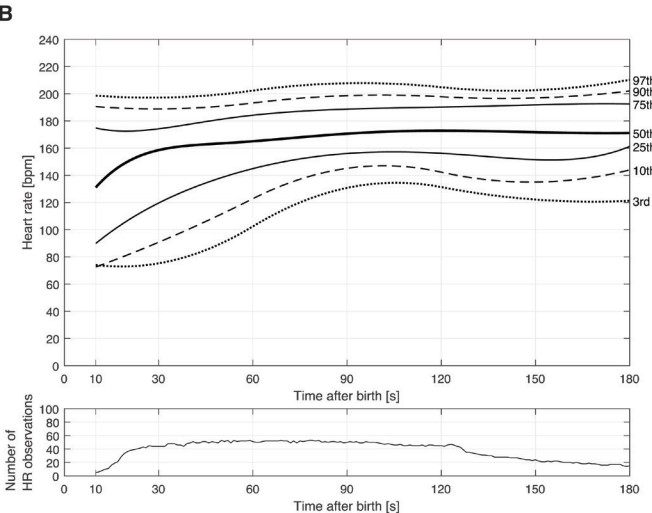

**Figure 2** (A) Smoothed heart rate percentiles for crying infants during the first 180 s after birth. (B) Smoothed heart rate percentiles for non-crying but breathing infants during the first 180 s after birth.

**Table 3** Adjusted OR for experiencing bradycardia (heart rate <100 beats/min) and tachycardia (heart rate ≥200 beats/min) after cord clamping within the first 180 s after birth

| | Bradycardia aOR (95% CI) | Tachycardia aOR (95% CI) |
|---|---|---|
| Non-crying but breathing | **2.64 (1.34 to 5.17, p=0.005)** | **2.86 (1.50 to 5.47, p=0.002)** |
| Gestational age (per week) | 0.98 (0.85 to 1.13, p=0.77) | 1.06 (0.92 to 1.22, p=0.41) |
| Birth weight (per kg) | 1.23 (0.77 to 1.99, p=0.38) | 1.09 (0.69 to 1.73, p=0.71) |
| Obstetric complications | 1.43 (0.73 to 2.80, p=0.30) | **3.43 (2.00 to 5.87, p<0.001)** |
| Early cord clamping (<60 s) | 1.12 (0.77 to 1.64, p=0.55) | 1.03 (0.72 to 1.48, p=0.86) |

Figures in bold indicate statistically significant difference at p>0.05.

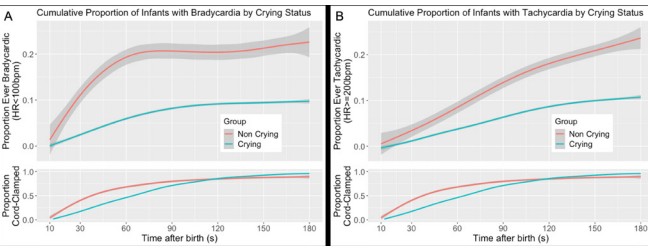

**Figure 3** Cumulative proportion of neonates with (A) bradycardia (heart rate <100 beats/min for ≥1 s) and (B) tachycardia (heart rate ≥200 beats/min for ≥1 s) among non-crying but breathing and crying groups during first 180 s after birth with 95% CI. Data were sampled every second and curves were generated using a generalised additive model.

Results of this study have potential implications for neonatal resuscitation algorithms and clinical practice. Current resuscitation guidelines are inconsistent in whether breathing or crying should guide the need for resuscitation (online supplemental figure 1). The HBB algorithm uses crying as an indicator that the infant is breathing well enough to not need stimulation.[2] In contrast, to avoid unnecessary handling, the recently released WHO Essential Newborn Care 1 algorithm suggests that if an infant is either crying or breathing well, then stimulation is not required.[20]

Differentiating well breathing from less effective respiratory effort is challenging, and saturation measurement or abdominal breathing monitoring is not always available in low-income settings. Findings of this study, along with previous research showing higher odds of mortality among non-crying but breathing infants as compared with those who were crying at birth,[10] suggest that the role of stimulation in non-crying but breathing neonates should be examined in a randomised trial. However, such study would be hard to conduct as the proportion of non-crying but breathing infants is low.

### Strengths

Outcome data of the study were measured with NeoBeat HR monitor that has been shown to record efficient and reliable HR data.[13] Trained external data collectors directly observed all births and classified crying status immediately after stabilisation.[11 12 21] To further increase reliability of the crying classification, we cross-tabulated data from breathing observations made real time during the initial stabilisation and found high rates of agreement.

### Limitations

Evaluating breathing patterns in a newborn can be difficult, and complex systems using flow measurement, video and audio recordings are required for confident classification.[5] In this study, crying and breathing status at birth were determined based on overall impression during the first 30 s after birth with unclear interobserver reliability and risk of bias compared with video recording. Any potential misclassification might nonetheless have created spurious associations as the number of non-crying but breathing infants was relatively small. The results should therefore be considered preliminary and advocate for validation with a video review. We were also unable to separate any form of breathing from breathing well, which might have biased the non-crying but breathing cohort towards sicker babies than what is intended with the Essential Newborn Care 1 resuscitation algorithm.[20] The same classification issue is, however, encountered by practitioners trying to apply the algorithm to clinical practice. Exclusion of circa 15% of babies born with caesarean section limits the generalisability of the findings.[12] This study was underpowered to explore differences in outcome beyond admission to neonatal intensive care unit. Lastly, the potential relationship between non-crying and hypoxia as the explanatory mechanism for bradycardia and tachycardia could not be defined due to lack of concomitant oxygen saturation measurement. Further studies allowing recorded review for classification of both crying and breathing status, coupled with simultaneous HR and saturation measurement, should be performed in a sufficiently large population group to provide definitive guidance.

## CONCLUSIONS

Crying is a simple clinical indicator of likely adequate lung recruitment facilitating normal physiological transition at birth. Infants who are breathing but *do not cry* are at increased risk of cardiopulmonary instability and should therefore be monitored closely.

**Author affiliations**
[1]Department of Women's and Children's Health, Uppsala University, Uppsala, Sweden
[2]Department of Pediatrics, Region Gävleborg, Gävle, Sweden
[3]Division of Neonatology, Department of Pediatrics, University of Washington, Seattle, Washington, USA
[4]Strategic Research, Laerdal Global Health, Stavanger, Rogaland, Norway
[5]Center on Human Development and Disability, University of Washington, Seattle, Washington, USA
[6]Golden Community, Lalitpur, Nepal
[7]School of Public Health and Community Medicine, Institute of Medicine, Sahlgrenska Academy, University of Gothenburg, Gothenburg, Sweden

**Acknowledgements** We thank the research nurses, paediatricians and nurses in the labor and delivery unit at Pokhara Academy of Health Sciences, Ram Chandra Bastola and Pratiksha Bhattarai for the clinical advice, and Nick Brown for his comments on the manuscript.

**Contributors** AKC designed the study; implemented, supervised and carried out the study and the data collection on site; reviewed the data analysis; and drafted the initial manuscript. He is responsible for the overall content as the guarantor and accepts full responsibility for the finished work and the conduct of the study, had access to the data, and controlled the decision to publish. SKB designed the study, reviewed data analysis and revised and edited the manuscript. AJK designed the study, reviewed the data analysis and drafted the initial manuscript. JE and TRW extracted and analysed the heart rate data and drafted the initial manuscript. OB implemented, supervised and carried out the study and the data collection on site, and extracted and analysed the heart rate data. All authors approved the final manuscript as submitted and agreed to be accountable for all aspects of the work.

**Funding** Laerdal Foundation for Acute Medicine, Norway (2019-40499) and Grand Challenges Canada (1910-30925).

**Competing interests** JE is employed at Laerdal Medical. AJK received PhD salary from Laerdal Foundation through a grant paid to Uppsala University.

**Patient and public involvement**  Patients and the public were not involved in the design, conduct, reporting or dissemination plans of this research.

**Patient consent for publication**  Not applicable.

**Ethics approval**  This study involves human participants and was approved by Nepal Health Research Council (number 87/2018). The participants gave informed consent to participate in the study before taking part. Written informed consent obtained from the pregnant women before delivery. Declaration of Helsinki by the World Medical Association was adhered to.

**Provenance and peer review**  Not commissioned; externally peer reviewed.

**Data availability statement**  Data are available upon reasonable request. Background data provided as online supplemental file.

**Open access**  This is an open access article distributed in accordance with the Creative Commons Attribution 4.0 Unported (CC BY 4.0) license, which permits others to copy, redistribute, remix, transform and build upon this work for any purpose, provided the original work is properly cited, a link to the licence is given, and indication of whether changes were made. See: https://creativecommons.org/licenses/by/4.0/.

**ORCID iDs**
Antti Juhani Kukka http://orcid.org/0000-0002-5879-7417
Joar Eilevstjønn http://orcid.org/0000-0002-4607-2689
Ashish KC http://orcid.org/0000-0002-0541-4486

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
