## [Reviewer comments · BMJ Paediatrics Open]

ARTICLE DETAILS

TITLE (PROVISIONAL)	Observational study comparing heart rate in crying and non-crying but breathing infants at birth
AUTHORS	Kukka, Antti Berkelhamer, Sara Eilevstjønn, Joar Wood, Thomas Basnet, Omkar KC, Ashish

VERSION 1 - REVIEW

REVIEWER	Reviewer Name: Dr. Peter Flom Institution and Country: Peter Flom Consulting 515 West End Ave New York 10024 United States
REVIEW RETURNED	03-Feb-2023

GENERAL COMMENTS	I confine my remarks to statistical and methodological aspects of this paper. These were very well done and I recommend publication.
---

REVIEWER	Reviewer Name: Dr. J Dekker Institution and Country: Leiden University Medical Centre Albinusdreef 2 Leiden 2333 ZA Netherlands
REVIEW RETURNED	17-Feb-2023

GENERAL COMMENTS	I would like to congratulate the authors on this well described, interesting paper. I have some minor suggestions: - Introduction: I don't quite understand what you mean with sentence 80-83. At first you describe the problem of partial lung aeration, but then you describe that despite that, PVR decreases resulting in better flow and preload for left ventricle. So you start this paragraph with saying that failure to rapidly recruit lung tissue can be an issue, but you explain the opposite. - Sentence 86-88: I don't think you can say that vigorous infants maintain CO solely by an increase in heartrate and thereby implicate that preload of the left ventricle does not play a role in this. The reference you mention did indeed not show a reduction in bradycardia in infants with PBCC, but infants in the DCC group cried earlier in life and needed less interventions for resuscitation. - Could you mention which time intervals you are comparing in the methods section (line 150)? - Lines 163-164: why would you mention this? - First paragraph of the results: the numbers do not correspond with figure 1, could you please check these?
---

	- Line 196: it looks like crying and non-crying are switched.- Regarding the second paragraph of the discussion: I wondered whether it could be possible that the non-crying babies were clamped early because of a reflex of the caregiver? And could that have influenced the outcomes?
--	--

VERSION 1 – AUTHOR RESPONSE

Dear Reviewers,

Thank you for the thoughtful review. We have responded to your queries in the attached document and modified the manuscript.

Antti Kukka

Reviewer: 1

Dr. Peter Flom, Peter Flom Consulting

Comments to the Author

I confine my remarks to statistical and methodological aspects of this paper.

These were very well done and I recommend publication.

Peter Flom

We would like to thank the reviewer for the recommendation.

Reviewer: 2

Dr. J Dekker, Leiden University Medical Centre

Comments to the Author

I would like to congratulate the authors on this well described, interesting paper. I have some minor suggestions:

We would like to thank the reviewer for the comment.

- Introduction: I don't quite understand what you mean with sentence 80-83. At first you describe the problem of partial lung aeration, but then you describe that despite that, PVR decreases resulting in better flow and preload for left ventricle. So you start this paragraph with saying that failure to rapidly recruit lung tissue can be an issue, but you explain the opposite.

Thank you for the helpful suggestion. We have reworded this paragraph to emphasize the physiologic response with lung recruitment specifically:

Lung recruitment at birth also plays a key role in cardiovascular transition and pulmonary blood flow. Lung aeration increases arterial oxygen content with subsequent decrease in local pulmonary vascular resistance, while shift of liquid from the alveoli to perialveolar tissue triggers a neurally mediated global increase in pulmonary blood flow [7]. This transition is key to supporting cardiac output with increases in pulmonary venous return.

Please see response below.

- Sentence 86-88: I don't think you can say that vigorous infants maintain CO solely by an increase in heartrate and thereby implicate that preload of the left ventricle does not play a role in this. The reference you mention did indeed not show a reduction in bradycardia in infants with PBCC, but infants in the DCC group cried earlier in life and needed less interventions for resuscitation.

Thank you again for the requested clarification. We have revised this paragraph to acknowledge your concerns as followed:

With clamping of the umbilical cord and closure of right-to-left shunts, pulmonary venous return becomes the sole source of left intracardiac volume [7]. If the cord is clamped before the lung is aerated, clinical compromise and a fall in cardiac output can theoretically lead to bradycardia [4,8]. However, a trial of physiologically-based cord clamping following established lung aeration did not show a reduction in the rate of bradycardia in infants needing resuscitation when compared to early cord clamping [9].

- Could you mention which time intervals you are comparing in the methods section (line 150)?

Thank you for the suggestion. Added mentioning of the intervals as below. The results are shown in Supplementary Table 2.

Median HRs at two serial time points were compared every 15 seconds using Mann Whitney U test with post hoc Bonferroni correction of p-values to adjust for repeated measures ($p=0.05/12 = p\leq 0.004$ for significance).

- Lines 163-164: why would you mention this?

It is the policy of BMJ journals that every article should have a patient involvement statement regardless of whether they were consulted in the process or not.

- First paragraph of the results: the numbers do not correspond with figure 1, could you please check these?

We thank the reviewer for pointing out this inconsistency. Late in the process of writing the manuscript, the Figure 1 was revised for better readability, but changes were not carried over to the text. Further inconsistencies in the tally of excluded cases appears in the Figure itself, which too has now been revised. The corrected version reads:

During the study period, 3578 deliveries took place at the hospital of which 1028 were not assessed for eligibility. Of the 2550 births observed, Liveborn was not used in 368 neonates, 57 resulted in a fresh stillbirth and 34 neonates had a gestational age less than 33 weeks. Of the neonates observed with Liveborn, 882 were excluded due to issues with the HR data, LiveBorn application or gestational age. Among the 1209 neonates included in the final analyses, 1155 were classified as crying and 54 as non-crying but breathing immediately after birth (Fig. 1).

- Line 196: it looks like crying and non-crying are switched.

Thank you for pointing out the error. Corrected version below:

A total of 6/54 (11.1%) and 48/1155 (4.2%) infants were ever bradycardic, and 6/54 (11.1%) and 71/1155 (6.1%) ever tachycardic in the non-crying and crying groups, respectively.

- Regarding the second paragraph of the discussion: I wondered whether it could be possible that the non-crying babies were clamped early because of a reflex of the caregiver? And could that have influenced the outcomes?

We much agree with the reviewer and this concern. As the study was not randomized, it is indeed possible that both our exposure variable and outcome variable might have impacted the clinicians' decision to clamp the cord, which in turn might have led to changes in HR. We suggest following addition:

Timing of cord clamping also impacts HR with early clamping increasing the risk of bradycardia [16,17]. In this study, non-crying but breathing neonates were more likely to undergo early cord clamping than crying neonates (75.9% vs 46.5%), but the difference in odds of bradycardia remained statistically significant after this and other adjustments. Early cord clamping may, however, explain some of the differences in timing of bradycardia observed. Furthermore, the clinicians' decision to perform early cord clamping might have been impacted by the lack of early cry or the observed HR.